American Society for Microbiology | Microbiology Spectrum

# Novel application of ribonucleoprotein-mediated CRISPR-Cas9 gene editing in plant pathogenic oomycete species

Erika N. Dort,[1] Nicolas Feau,[2] Richard C. Hamelin[1,3,4]

**ABSTRACT** CRISPR-Cas9 gene editing has become an important tool for the study of plant pathogens, allowing researchers to functionally characterize specific genes involved in phytopathogenicity, virulence, and fungicide resistance. Protocols for CRISPR-Cas9 gene editing have already been developed for Phytophthoras, an important group of oomycete plant pathogens; however, these efforts have exclusively focused on agricultural pathosystems, with research lacking for forest pathosystems. We sought to develop CRISPR-Cas9 gene editing in two forest pathogenic Phytophthoras, *Phytophthora cactorum* and *P. ramorum*, using a plasmid-ribonucleoprotein (RNP) co-transformation approach. Our gene target in both species was the ortholog of *PcORP1*, which encodes an oxysterol-binding protein that is the target of the fungicide oxathiapiprolin in the agricultural pathogen *P. capsici*. We delivered liposome complexes, each containing plasmid DNA and CRISPR-Cas9 RNPs, to *Phytophthora* protoplasts using a polyethylene glycol-mediated transformation protocol. We obtained two *ORP1* mutants in *P. cactorum* but were unable to obtain any mutants in *P. ramorum*. The two *P. cactorum* mutants exhibited decreased resistance to oxathiapiprolin, as measured by their radial growth relative to wild-type cultures on oxathiapiprolin-supplemented medium. Our results demonstrate the potential for RNP-mediated CRISPR-Cas9 gene editing in *P. cactorum* and provide a foundation for future optimization of our protocol in other forest pathogenic *Phytophthora* species.

**IMPORTANCE** CRISPR-Cas9 gene editing has become a valuable tool for characterizing the genetics driving virulence and pathogenicity in plant pathogens. CRISPR-Cas9 protocols are now well-established in several *Phytophthora* species, an oomycete genus with significant economic and ecological impact globally. These protocols, however, have been developed for agricultural *Phytophthora* pathogens only; CRISPR-Cas9 systems have not yet been developed for any forest pathogenic Phytophthoras. In this study, we sought to establish CRISPR-Cas9 gene editing in two forest *Phytophthora* pathogens that cause widespread tree mortality: *P. cactorum* and *P. ramorum*. We successfully obtained gene mutations in *P. cactorum* and demonstrated a decrease in fungicide resistance, a trait that could impact the pathogen's ability to cause disease. However, the same protocol did not yield any mutants in *P. ramorum*. The results of our study will serve as a baseline for the development of CRISPR-Cas9 gene editing in forest Phytophthoras and other oomycetes.

**KEYWORDS** *Phytophthora*, gene disruption, plasmid co-transformation, forest pathology, antifungal resistance

The oomycete genus *Phytophthora* comprises some of the most ecologically and economically impactful filamentous plant pathogens in agriculture, forestry, and horticulture. Well-known examples include *P. infestans*, the causal agent of late blight of potato (1), and *P. cinnamomi*, a devastating plant pathogen with thousands of host

Address correspondence to Erika N. Dort, edort@alum.ubc.ca.

The authors declare no conflict of interest.

See the funding table on p. 16.

species and worldwide distribution (2), widely known as the causal agent of jarrah dieback in Australia (3). Due to the global impacts of Phytophthoras on plant health, characterizing the genetic mechanisms driving the phytopathogenicity of this genus is a research priority (4–6). Since its development in 2012 (7), CRISPR-Cas9 gene editing has become a valuable tool for this molecular characterization, facilitating exploration of the genes underlying host-*Phytophthora* interactions (8, 9) and elucidating potential gene targets for disease resistance and control strategies (10, 11).

CRISPR-Cas gene editing technologies are based on the naturally occurring adaptive immune systems of bacteria and archaea: clustered regularly interspaced short palindromic repeats (CRISPR) and CRISPR-associated (Cas) proteins (12–14). CRISPR-Cas9 gene editing was developed in 2012 by Jinek et al. (7) by adapting the type II CRISPR-Cas system of the bacterium *Streptococcus pyogenes*. In their 2012 study, Jinek et al. demonstrated that type II CRISPR-Cas systems target double-stranded DNA with three components: a Cas9 endonuclease (Cas9), a CRISPR RNA (crRNA), and a trans-activating CRISPR RNA (tracrRNA). The crRNA and tracrRNA base-pair together and bind to Cas9, and a 20-nucleotide (nt) region of the crRNA guides Cas9 to the complementary DNA in the target organism's genome, called the protospacer, where Cas9 creates a double-stranded break (DSB) near a highly conserved sequence (NGG) called the protospacer-adjacent motif, or PAM. Jinek et al. (7) further demonstrated that the CRISPR-Cas9 system could be modified to become a highly adaptable gene editing technology by fusing the crRNA and tracrRNA into a single-guide RNA (sgRNA); the sgRNA can direct Cas9 to cleave virtually any protospacer target near an NGG PAM via modifications made to the 20-nt crRNA region. In eukaryotes, the DSB is then repaired by the host cell, either through an error-prone end-joining pathway in the absence of a homologous donor template, or through the highly accurate homology-directed repair pathway in the presence of a homologous donor template (15). It is through the DSB repair process that mutations are made in the target gene, which allows researchers to characterize gene functions via knockout or highly specific mutations.

The CRISPR-Cas9 systems developed in filamentous (fungal and oomycete) plant pathogens have enabled researchers to discover how specific genes contribute to phytopathogenicity (16, 17), elucidate important secondary metabolite and toxin pathways (18, 19), and characterize genetic resistance to fungicides (20, 21). To date, however, most CRISPR-Cas9 research in plant pathology has focused on agricultural pathosystems, with very little research directed toward forest pathogens. For example, the recent development of a CRISPR-Cas9 system in the fungal pathogen *Dothistroma septosporum* is the first use of CRISPR-based gene editing in a filamentous forest pathogen (22). CRISPR-Cas9 gene editing protocols have now been developed for several agricultural oomycete pathogens including *P. sojae* (16), *P. palmivora* (8), *P. capsici* (21, 23), *P. litchii* (24), *P. parasitica* (25), and *P. colocasiae* (26). These studies all used polyethylene glycol (PEG)-mediated transformation to deliver the CRISPR-Cas9 components to *Phytophthora* protoplasts, aside from Gumtow et al., who used *Agrobacterium*-mediated transformation (AMT) to deliver the Cas9 and sgRNA to *P. palmivora* (8). Despite the success of CRISPR-Cas9 systems in agricultural Phytophthoras, there have been no such methods developed for forest pathogenic Phytophthoras, which may be attributed to a lack of efficient and stable transformation protocols in these species (27).

When delivering CRISPR-Cas9 components to cells via PEG-mediated transformation, two general methods can be employed. The first is plasmid delivery, where both Cas9 and its sgRNA are encoded on a plasmid that is delivered through transformation to the nucleus and expressed in the cell via transcription and translation. The translated Cas9 nuclease and sgRNA then form a ribonucleoprotein (RNP) complex in the cell and return to the nucleus to find the target protospacer DNA and create a DSB. The second method is the direct transformation of a purified Cas9-sgRNA RNP complex, which also gets delivered to the nucleus through transformation, but does not require transcription and translation and thus immediately begins targeting the genomic protospacer DNA. First developed in the nematode *Caenorhabditis elegans* (28), the RNP method for

CRISPR-Cas9 gene editing has been successfully employed in mammals (29), plants (30), and fungi (31). While RNP delivery of Cas9 has not been developed for any *Phytophthora* species, it has been used to successfully edit the genome of the oomycete *Aphanomyces invadans* (32).

RNP delivery of CRISPR-Cas9 components has several advantages over plasmid delivery. First, it is possible to obtain CRISPR-Cas9 mutants with no transgenic DNA, which has motivated RNP method development in plants (30, 33). Another advantage of RNP delivery is that the RNP complexes can target DNA immediately after transformation into cells, as opposed to plasmid delivery which requires time for transcription and translation (28). In fact, the transgenic expression of Cas9 can have toxic effects in many organisms (34–37), including oomycetes (38, 39), which can usually be avoided with the transient expression of RNPs (35). Finally, the use of RNPs for CRISPR-Cas9 delivery avoids many of the common challenges of plasmid DNA transformation, including potential off-target effects of random transgene insertion into genomes (29, 40, 41) and the need to optimize plasmid gene promoters and codon usage for each target organism (28, 30). One significant limitation of the RNP approach, however, is the lack of a selective marker for putative transformants, particularly when triggering the end-joining DNA repair pathway with no donor template. This lack of selection is particularly problematic when the target gene for CRISPR-Cas9 produces no visible phenotype when mutated.

The objective of this study was to establish CRISPR-Cas9 gene editing using an RNP-mediated approach in two forest pathogenic Phytophthoras: *Phytophthora ramorum* Werres, De Cock, & Man in't Veld and *Phytophthora cactorum* (Leb. & Cohn) Schröeter. *Phytophthora ramorum*, the causal agent of sudden oak death and sudden larch death, is a globally invasive pathogen causing landscape-level damage in the tanoak/oak forests of California (42, 43) and larch plantations in the United Kingdom (44). *Phytophthora cactorum* is an aggressive pathogen with a broad host range spanning hundreds of agricultural, horticultural, and forest species (45–47). As a forest tree pathogen, *P. cactorum* is perhaps most destructive in nurseries (48–50), but it has also been associated with diseases on economically and ecologically important genera in forest stands including beech (51), ash (52), pine (53), and oak (54). The gene we chose to target is the ortholog of *PcORP1* in *P. capsici*, named for the oxysterol-binding protein (OSBP) that it encodes, which is a member of the OSBP-related proteins (ORPs) family (55). The ORP1 protein in *Phytophthora* species is the target of the fungicide oxathiapiprolin (56–58), and mutations in the *ORP1* gene of *P. capsici*, *P. colocasiae*, *P. nicotianae*, and *P. sojae* have yielded oxathiapiprolin-resistant mutants (21, 26, 55, 59). We tested a co-transformation approach for CRISPR-Cas9 editing in *P. ramorum* and *P. cactorum*, delivering liposomes containing both a Cas9-sgRNA RNP targeting the *ORP1* gene and a plasmid encoding the *nptII* gene for geneticin (G418) resistance. We chose this hybrid approach to facilitate the selection of putative mutants, hypothesizing that *Phytophthora* transformants growing on G418-selective medium would have received both the plasmid DNA and the CRISPR-Cas9 RNP complexes and that the mutant screening process would therefore be more efficient than without a selection method.

## MATERIALS AND METHODS

The detailed recipes and protocols for the V8, nutrient V8, and V8-mannitol growth media and the W5, MMg, PEG, and enzyme solutions are available in Dort and Hamelin (27). All PCRs in this study were performed using a Bio-Rad T100™ thermal cycler (Hercules, CA, USA).

### *Phytophthora* isolates and culture conditions

Cultures of *P. cactorum* (isolate Larch FF-42 2Pa [abbreviated FF42]; [60]) and *P. ramorum* (isolate NA2 17_0134_0030 [abbreviated NA2_17]; G. Bilodeau, *pers. com.*) were maintained on clarified 20% V8 agar (V8A) at ambient temperature (approx. 21°C) in the

dark. For long-term storage, agar plugs taken from the edge of mycelial growth, 5 days post-plating, were stored in nuclease-free molecular water at 5–10°C in the dark.

## *Phytophthora* DNA extractions

Agar plugs were taken from the edge of mycelial growth of *Phytophthora* cultures and transferred to a 12-well plate, with each well containing one agar plug in 2 mL of clarified 20% V8 broth (V8B). After 3–4 days of growth in V8B, 50–100 mg of mycelial tissue was harvested from each culture and flash frozen in liquid nitrogen. Tissue disruption was performed using a Retsch Mixer Mill MM 300 (Retsch GmbH, Haan, Germany). Total cellular DNA was extracted using a QIAGEN DNeasy Plant Mini Kit (QIAGEN Sciences, Germantown, MD, USA) as per the manufacturer's instructions (DNeasy Plant Handbook, Mini Protocol, TissueLyser procedure).

## PCR primer design

All PCR primers used in this study were designed using NCBI Primer-BLAST software (61) with target specificity checked against the NCBI "RefSeq representative genomes" database for *Phytophthora* (taxid: 4783) to minimize off-target amplification. Primers were synthesized by Integrated DNA Technologies (IDT, Coralville, IA, USA).

## Finding the *PcORP1* orthologs in *P. ramorum* and *P. cactorum*

The OSBP protein target of oxathiapiprolin was first identified in *P. capsici* (56, 58) and was labeled in the corresponding JGI PhycoCosm (62) genome (*Phytophthora capsici* LT1534 v11.0) as Protein Id: 564296 (PHYCAscaffold_14:545019–548397). The gene for the *P. capsici* OSBP was first referred to as *PcORP1* (55); we chose to follow this naming convention in our species, referring to the orthologs in *P. cactorum* and *P. ramorum* as *PcaORP1* and *PrORP1*, respectively. The detailed methods for the identification, sequencing, and gene prediction of the *PcORP1* orthologs in *P. ramorum* and *P. cactorum* are provided in the Supplementary Methods (File S1).

## CRISPR-Cas9 guide RNA (gRNA) design

gRNAs were designed within the *PrORP1* and *PcaORP1* predicted ORF regions. Intron regions of each ORF were avoided. The Eukaryotic Pathogen CRISPR Guide RNA/DNA Design Tool (EuPaGDT; [63]) was used to generate gRNAs with the SpCas9 20-nucleotide gRNA 3′ NGG PAM option selected (off-target PAM motif setting: NAG, NGA). Off-target analysis was performed in EuPaGDT using the *P. ramorum* Pr-102 FungiDB-26 genome (NCBI accession GCA_020800215.1) for the *PrORP1* gRNAs and the *P. cactorum* P414 NCBI reference genome (NCBI accession GCA_016864655.1) for the *PcaORP1* gRNAs. Note that when the first three PcaORP1 gRNAs were designed, the *P. cactorum* P414 genome was not provided in EuPaGDT, so off-target analysis was performed manually using a blastn (64) search of each gRNA candidate against the P414 genome in NCBI. Once the P414 genome was added, the subsequent two gRNAs for PcaORP1 were designed using the off-target analysis in EuPaGDT. All gRNAs with a EuPaGDT "Total score" above 0.5 (as per [65]) and no off-target hits were considered potential candidates, and their secondary structure was analyzed using RNAstructure (66). Only gRNAs with fewer than three consecutive paired bases were considered for further analysis (65). Finally, candidate gRNAs were analyzed with the Integrated DNA Technologies (IDT) "CRISPR-Cas9 guide RNA design checker"; IDT does not provide a cutoff score for gRNAs, but states that a higher score may indicate better on-target performance of the gRNA. Therefore, the final gRNAs chosen for this study were those with the highest possible IDT scores combined with minimal secondary structure (as per the RNAstructure analysis). All gRNAs were ordered as 2 nmol Alt-R CRISPR RNAs (crRNAs) from IDT (Coralville, IA, USA).

## RNP-plasmid co-transformation

The methods for preparation of *Phytophthora* cultures and protoplast isolation are based on the optimized transformation protocols developed for *Phytophthora* species (65, 67, 68) and tested in *P. cactorum* and *P. ramorum* previously (27). All centrifugation steps were carried out using an Eppendorf 5810R centrifuge with an A-4-62 rotor.

### Plasmid DNA

The pYF2-PsNLS-hSpCas9-GFP (abbreviated pYF2-PsCG) plasmid, previously demonstrated to generate geneticin (G418)-resistant transformants in *P. cactorum* and *P. ramorum* (27), was used for all co-transformations. Plasmid DNA was obtained using a QIAGEN HiSpeed Plasmid Maxi Kit (QIAGEN Sciences, Germantown, MD, USA) as per the manufacturer's instructions (HiSpeed Plasmid Purification Handbook), and vacuum concentrated to 3–5 µg/µL prior to use in co-transformations.

### Formation of Cas9 RNP complexes

The following components for the CRISPR-Cas9 RNP complexes were ordered from IDT (Coralville, IA, USA): Alt-R S.p. HiFi Cas9 Nuclease V3 (100 µg), Alt-R CRISPR-Cas9 crRNA (2 nmol), and Alt-R CRISPR-Cas9 tracrRNA (20 nmol). Prior to use, the Cas9 nuclease was diluted with W5 solution to a 1 µM working solution and stored at −80°C (as per instructions from IDT). Both the crRNA and tracrRNA were resuspended to make 100 µM stocks using IDT Nuclease-Free Duplex Buffer (NFDB). gRNAs were formed by creating 1 µM crRNA-tracrRNA duplexes as follows: equal volumes of 100 µM crRNA and 100 µM tracrRNA were mixed in a 1.5 mL microcentrifuge tube with NFDB for a final concentration of 1 µM, incubated at 95°C for 5 min, cooled to ambient temperature, and stored at −20°C. The Cas9-gRNA RNP complexes were formed as follows: equal volumes of 1 µM gRNA and 1 µM Cas9 nuclease were mixed in a 1.5 mL microcentrifuge tube with W5 solution for a final concentration of 300 nM, incubated at ambient temperature for 5 min, and stored at 4°C until used for transformation. For transformations with Lipofectamine CRISPRMAX (Thermo Fisher Scientific, Waltham, MA, USA), Cas9 PLUS Reagent was added to the RNP mixture at a ratio of 0.6 µL of reagent to 25 µL total RNP volume (as per IDT Alt-R CRISPR-Cas9 User Guide, protocol for cationic delivery of Cas9 RNP).

### Preparation of Phytophthora cultures for transformation

*Phytophthora* cultures were initially grown on 1.5% nutrient V8 agar (NVA) plates at ambient temperature (approx. 21°C) in the dark. Once there was sufficient mycelial growth (5–7 days), cultures were transferred to nutrient V8 broth (NVB): three flasks, each containing 50 mL of NVB, were inoculated with five agar plugs (6 mm diameter) taken from the growing edge of the mycelial culture (15 agar plugs total). The liquid cultures were grown for 3–4 days at ambient temperature (approx. 21°C) in the dark.

### Isolation of protoplasts

After 3 days of growth, mycelial mats were harvested and rinsed with autoclaved Milli-Q water followed by 0.8 M mannitol. The mycelial mats were transferred to a 100 mm × 20 mm Petri dish, covered with 0.8 M mannitol, and shaken gently (approx. 30 rpm) for 10 min. The mats were recollected, transferred to a new Petri dish containing 20 mL of an enzyme solution (lysing enzymes + cellulase) and digested for 60–90 min at 25°C with gentle shaking (approx. 30 rpm). After digestion, the enzyme/protoplast solution was filtered through a 70 µm cell strainer and centrifuged at 1,200 rcf for 2.5 min. The pellet was gently washed in 35 mL of W5 solution, re-centrifuged at 1,200 rcf for 3 min, and the resulting pellet was resuspended in 10 mL of W5 solution and placed on ice for 1–2 h. The protoplast concentration was determined using a hemocytometer.

## Formation of plasmid-RNP cationic-lipid transfection complexes (TCs)

While the protoplasts were incubating on ice, plasmid-RNP TCs were prepared as follows: 250 µL of 300 nM RNP, 6 µL of pYF2-PsCG plasmid DNA, 12 µL of Lipofectamine CRISPRMAX or Lipofectamine RNAiMAX (Thermo Fisher Scientific, Waltham, MA, USA), and 232 µL of W5 solution were mixed in a 1.5 mL microcentrifuge tube and incubated at ambient temperature for 20 min. The volumes of RNP and Lipofectamine were determined based on the picomoles of RNP per cell used in the IDT protocol for cationic lipid delivery of CRISPR RNP complexes (69). The final TC for each transformation had a total volume of 500 µL containing 75 pmol of RNP and approx. 33 µg (4 pmol) of plasmid DNA.

## Co-transformation of Phytophthora protoplasts with CRISPR-Cas9 RNP complexes and plasmid DNA

Multiple protocols were tested for plasmid-RNP co-transformations in *P. ramorum* and *P. cactorum*; the details of the methods tested are summarized in Data S1. The protocol outlined below is the method that ultimately succeeded in *P. cactorum* and is a modified version of the plasmid-mediated CRISPR-Cas9 transformation developed for *P. sojae* (65).

Following incubation on ice, the protoplast suspension was centrifuged at 1,200 rcf for 4 min. The resulting pellet was resuspended in MMg solution at a concentration of approx. $1.0-3.0 \times 10^6$ protoplasts/mL and incubated at ambient temperature for 10 min. During incubation, 500 µL of plasmid-RNP TC, or 500 µL of sterile molecular-grade water/W5 solution for the negative control, was added to a 50 mL centrifuge tube and placed on ice. After the 10-min incubation, the protoplasts were gently mixed, and 1 mL of the protoplast/MMg solution was added to the TC (or control) in each 50 mL tube, mixed, and incubated for 30 min on ice. A 40% (vol/vol) PEG solution was added to each tube in three aliquots of 580 µL for a total of 1.74 mL of PEG solution per tube, and the tubes were incubated for another 30 min on ice. After 30 min, 2 mL of V8-mannitol (VM) broth was added to each tube. The tubes were inverted to mix and incubated on ice for 2 min. Then, 8 mL of VM broth was added to each tube, the tubes were inverted to mix, and incubated at ambient temperature for an additional 2 min. To test the viability of the protoplasts after the transformation process, approx. 30–50 µL of the protoplast-VM solution from each centrifuge tube (including control treatment) was spread onto an agar plate (NVA) and incubated at ambient temperature in the dark overnight (protoplast viability test Day 1). The remainder of the protoplast-VM solution (approx. 10 mL) from each tube was added to a 100 mm × 20 mm Petri dish containing 10 mL of VM broth with 100 µg/mL of ampicillin (Thermo Fisher Scientific, Waltham, MA, USA) for a final concentration of 50 µg/mL ampicillin. The dishes were incubated overnight at ambient temperature in the dark.

### Antibiotic selection of transformants

The following morning, the regenerated protoplasts were transferred to a new 50 mL tube and centrifuged at 2,000 rcf for 5 min. The pellet was resuspended in approx. 2 mL of the supernatant, and 150 µL of the resuspended protoplasts was added to 20 mL of cooled VM agar (40–45°C) without G418 and poured into a Petri dish (protoplast viability test Day 2). Cooled VM agar (40–45°C) with 30 µg/mL of G418 (Gibco Geneticin Selective Antibiotic, Thermo Fisher Scientific, Waltham, MA, USA) was added to the remaining protoplasts to a total volume of 50 mL. The tubes were inverted to mix, and the protoplast/agar mixture was poured into three Petri dishes (approx. 16–17 mL of mixture in each dish). All plates were incubated at ambient temperature in the dark for 2 days, or until mycelial growth appeared (up to 5 days). Each mycelial colony that grew on the G418-supplemented medium was treated as a putative transformant and was transferred to a well of a 12-well culture plate containing 2 mL of V8B with 40 µg/mL of G418.

## Screening transformants for CRISPR-Cas9 mutants

Any putative transformants that grew through the second round of G418 selection were considered successful transformants based on their stable growth on selective medium. Wild-type (WT) cultures were grown in parallel as a negative control. After 3–4 days of growth in the V8B + 40 µg/mL of G418 (no G418 for WT cultures), 50–100 mg of mycelial tissue was harvested from each culture and flash-frozen in liquid nitrogen. Tissue disruption was performed using a Retsch Mixer Mill MM 300 (Retsch GmbH, Haan, Germany). Total cellular DNA was extracted using a QIAGEN DNeasy Plant Mini Kit (QIAGEN Sciences, Germantown, MD, USA) as per the manufacturer's instructions (DNeasy Plant Handbook, Mini Protocol, TissueLyser procedure). PCR was performed on both transformant and WT DNA extractions using Phusion Plus DNA Polymerase (Thermo Fisher Scientific, Waltham, MA, USA) and primers designed to amplify the regions of *PrORP1* or *PcaORP1* targeted by the Cas9 gRNAs (Table S3). All PCRs were performed using the following thermocycler conditions (as per the manufacturer's instructions for Phusion polymerase): an initial hold of 98°C for 30 s; 35 cycles of 98°C for 10 s, 60°C for 10 s, 72°C for 30 s; and a final hold of 72°C for 5 min. The resulting PCR products were analyzed using gel electrophoresis and sent for Sanger sequencing in the forward direction. Sanger sequencing chromatograms were analyzed using the TIDE (Tracking of Indels by Decomposition) web tool to detect CRISPR-Cas9-generated insertions and deletions (indels) at the Cas9 cut site for each gRNA (70); the *P. cactorum PcaORP1* WT sequencing chromatogram was used as the control sample, and the *PcaORP1* mutant chromatograms were used as the test samples. Any identified *ORP1* mutations were then re-sequenced in the reverse direction and the sequence reads were re-analyzed with TIDE to confirm the mutation on both DNA strands.

Transformant cultures with *ORP1* mutations were taken through a single zoospore isolation protocol to ensure that downstream phenotyping would be performed on pure cultures derived from a mononucleate gene-edited zoospore. The detailed methods for the generation of single zoospore-derived cultures and the subsequent testing for plasmid DNA retention are provided in the Supplementary Methods (File S1).

## Oxathiapiprolin growth assays

As mutations were only detected in *PcaORP1*, fungicide phenotyping was performed only on *P. cactorum*. Oxathiapiprolin growth assays were performed to determine whether the CRISPR-Cas9 *PcaORP1* mutations would have any effect on the growth phenotype relative to WT *P. cactorum* cultures. Technical grade oxathiapiprolin (98.1%; Syngenta Canada Inc., Guelph, ON, Canada) was dissolved in DMSO to make a stock solution of 100 mg/mL. *P. cactorum* FF42 WT and *PcaORP1* mutant cultures were grown on V8A plates supplemented with a gradient of nine oxathiapiprolin concentrations: 0.0001, 0.0002, 0.0004, 0.0006, 0.0008, 0.001, 0.01, 0.1, and 1.0 µg/mL. The concentration of DMSO in the medium for each treatment was standardized to 0.1%, and the volume of V8A per plate was standardized to 15 mL. In parallel, the cultures were also grown on two control treatments: unamended V8A and V8A with 0.1% DMSO. Two individual single-zoospore-derived cultures were tested from the WT and each of the *PcaORP1* mutants, and four replicates were grown for each culture on each growth medium treatment. For each treatment, a 5 mm agar plug was taken from the edge of an actively growing 6-day-old culture and transferred to a treatment plate. Seven days after plating, the radial mycelial growth was measured in two perpendicular directions (Fig. S1); 5 mm was subtracted from each measurement to account for the diameter of the agar plug. The average of these two measurements was recorded as the radial growth for each culture.

## Statistical analyses

All statistical analyses were performed using Python version 3.8 with the following packages: pandas (71, 72), SciPy stats (73), scikit-posthocs (74), and statsmodels (75).

The resulting figures were also generated in Python 3.8 with the following packages: matplotlib (76), seaborn (77), pandas (71, 72), and statannotations (78).

The differences in average radial growth between *P. cactorum* FF42 WT and *PcaORP1* mutant cultures on oxathiapiprolin-supplemented V8A were analyzed using either a one-way ANOVA or Kruskal-Wallis test, depending on the normality of the data. The 0.1% DMSO control treatment was tested in two separate experiments, so the results from the two experiments were pooled. The results from each of the two individual SZCs for the WT and *PcaORP1* mutant cultures were also pooled (a total of eight replicates per isolate per treatment). A Shapiro-Wilk test of normality was performed on the growth data within each treatment. For treatments with normally distributed data, the differences in radial growth between isolates were analyzed with a one-way ANOVA followed by a Tukey's honest significance difference *post hoc* test. For treatments with non-normally distributed data, the differences in radial growth between isolates were analyzed with a Kruskal-Wallis test followed by a Dunn's multiple comparison *post hoc* test.

## RESULTS

### Structure of the *PrORP1* and *PcaORP1* genes

The *PrORP1* gene from *P. ramorum* NA2_17 was predicted to be 3,235 bp long, with two exons separated by a 76 bp intron (Fig. 1A; Data S2). The predicted 2,745 bp *PrORP1* ORF lies within the second exon and encodes a 914 amino acid polypeptide (Data S2). The *PcaORP1* gene from *P. cactorum* FF42 was predicted to be a total length of 3,166 bp, with three exons separated by two introns 82 and 96 bp in length (Fig. 1B; Data S2). The predicted 2,757 bp *PcaORP1* ORF spans the second and third exons and encodes an 886 amino acid polypeptide after intron splicing (Data S2).

Relative to the ORF of the *P. capsici ORP1* gene (*PcORP1*), the *PrORP1* and *PcaORP1* ORFs were predicted to have different lengths and exon/intron structures (Fig. S2A), but their InterPro-predicted protein domains were identical to PcORP1 (Fig. S2B). All three ORP1 proteins have a conserved structure: an N-terminal pleckstrin homology (PH) domain, a steroidogenic acute regulatory-related lipid-transfer (START) or START-like domain, and a C-terminal OSBP domain (Fig. S2B).

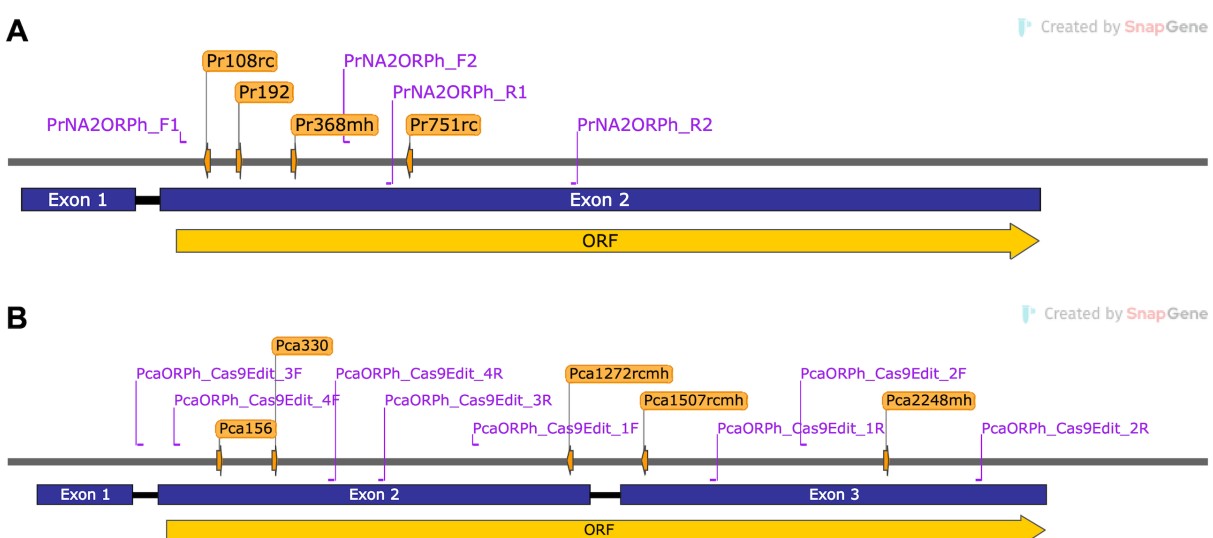

**FIG 1** Predicted structures of (A) the *PrORP1* gene from *Phytophthora ramorum* NA2_17, and (B) the *PcaORP1* gene from *Phytophthora cactorum* FF42. The orange labels indicate the locations of the guide RNAs designed for CRISPR-Cas9 gene editing (see Table 1), and the purple labels show the locations of the primer pairs designed to amplify the Cas9 ribonucleoprotein-targeted regions for Sanger sequencing (see Table S3). Gene map created using SnapGene Viewer software v.7.0.1 (from Dotmatics; available at snapgene.com).

## *PrORP1* and *PcaORP1* gRNAs

We tested four gRNAs for *PrORP1* and five gRNAs for *PcaORP1* for plasmid-RNP co-transformations in *P. ramorum* and *P. cactorum*, respectively. The four *PrORP1* gRNAs had EuPaGDT efficiency scores ranging from 0.58 to 0.82 (Table 1) and targeted regions within Exon 2 of *PrORP1* (Fig. 1A). The five *PcaORP1* gRNAs had EuPaGDT efficiency scores ranging from 0.55 to 0.64 (Table 1) and targeted regions within Exons 2 and 3 of *PcaORP1* (Fig. 1B).

## CRISPR-Cas9 plasmid-RNP co-transformation produces mutants in *P. cactorum* but not *P. ramorum*

### *P. cactorum*

Using a PEG-liposome-mediated plasmid-RNP co-transformation method (Fig. 2), we obtained two independent, heterozygous indel *PcaORP1* mutants in *P. cactorum* FF42. The first mutant, PT10, was obtained with the Pca1507rcmh gRNA, and TIDE analysis revealed a single base pair (bp) insertion between nucleotides 1,491 and 1,492 in the open reading frame (ORF) of the *PcaORP1* gene (*PcaORP1::1492*; Fig. 3). The TIDE results indicated that the nucleotide inserted on the sense strand of *PcaORP1* is a thymine; therefore, the nucleotide inserted on the antisense strand where the gRNA was designed is an adenine (Fig. 3). The second mutant, PT9, was obtained with the Pca156 gRNA, and TIDE analysis revealed a single base pair (bp) deletion of nucleotide 172 in the *PcaORP1* ORF (*PcaORP1Δ172*; Fig. 4). For both mutants, the ratio of WT to mutant DNA in the originally obtained transformant cultures was generally around 1.5–2:1, whereas the same ratio in single zoospore cultures (SZCs) shifted to 1:1, indicating that pure heterozygous cultures from a single-edited nucleus were obtained through the zoospore isolation process (Fig. 4 and 5; Fig. S3 and S4). Of note is that one of the SZCs obtained from the original PT9 mutant culture was no longer a *PcaORP1Δ172* mutant (PT9-1; Fig. 4B), indicating that it was derived from a zoospore with a non-edited WT nucleus. Both

**TABLE 1** CRISPR-Cas9 guide RNAs (gRNAs) designed to target the *PrORP1* and *PcaORP1* genes of *Phytophthora ramorum* and *P. cactorum*, respectively[b]

| Species (gene) | gRNA name (abbreviation) | PAM | 20-nt gRNA sequence (5′ to 3′) | EuPaGDT total score | IDT gRNA score |
|---|---|---|---|---|---|
| *Phytophthora ramorum* NA2_17 (*PrORP1*) | PrNA2ORPh_108rc (Pr108rc) | AGG | GTCCTTCCTCGAGATGTAAT | 0.58 | 45 |
| | PrNA2ORPh_192 (Pr192) | CGG | TTACCTCACGGACCCAGCTA | 0.61 | 51 |
| | PrNA2ORPh_368mh (Pr368mh) | CGG | ACACGAGGACACCGGCGCAG | 0.82 | 54 |
| | PrNA2ORPh_751rc (Pr751rc) | CGG | GACGGAACACAACGGAGTTG | 0.61 | 63 |
| *Phytophthora cactorum* FF42 (*PcaORP1*) | PcaFF42ORPh_156 (Pca156) | AGG | CTTGGAAGGTTGTACAGTGC | 0.64 | 57 |
| | PcaFF42ORPh_330 (Pca330) | AGG | CACAAGTGAGACCAATGCAC | 0.55 | 94 |
| | PcaFF42ORPh_1272rcmh[a] (Pca1272rcmh) | CGG | CACCACAGACTGGCCCATCT | 0.59 | 59 |
| | PcaFF42ORPh_1507rcmh[a] (Pca1507rcmh) | AGG | CCTGGTTCACGTATTCACTG | 0.57 | 73 |
| | PcaFF42ORPh_2248mh[a] (Pca2248mh) | TGG | CTGCCTTACTTACAGATCGG | 0.56 | 38 |

[a]gRNAs were designed before the *P. cactorum* P414 NCBI reference genome was added to EuPaGDT, so off-target analysis was performed manually using a blastn analysis in NCBI.
[b]All gRNAs are designed using a 3′ NGG protospacer adjacent motif (PAM). The number in each gRNA name represents the location of its first nucleotide in the *ORP1* open reading frame sequence. The gRNAs labeled with "rc" (reverse complement) are those derived from the antisense strand of the *ORP1* sequence. The gRNAs labeled with "mh" (microhomology) are those that EuPaGDT found to have microhomology pairs flanking the gRNA sequence. Also provided are the total scores from EuPaGDT for each gRNA and the IDT gRNA design checker scores.

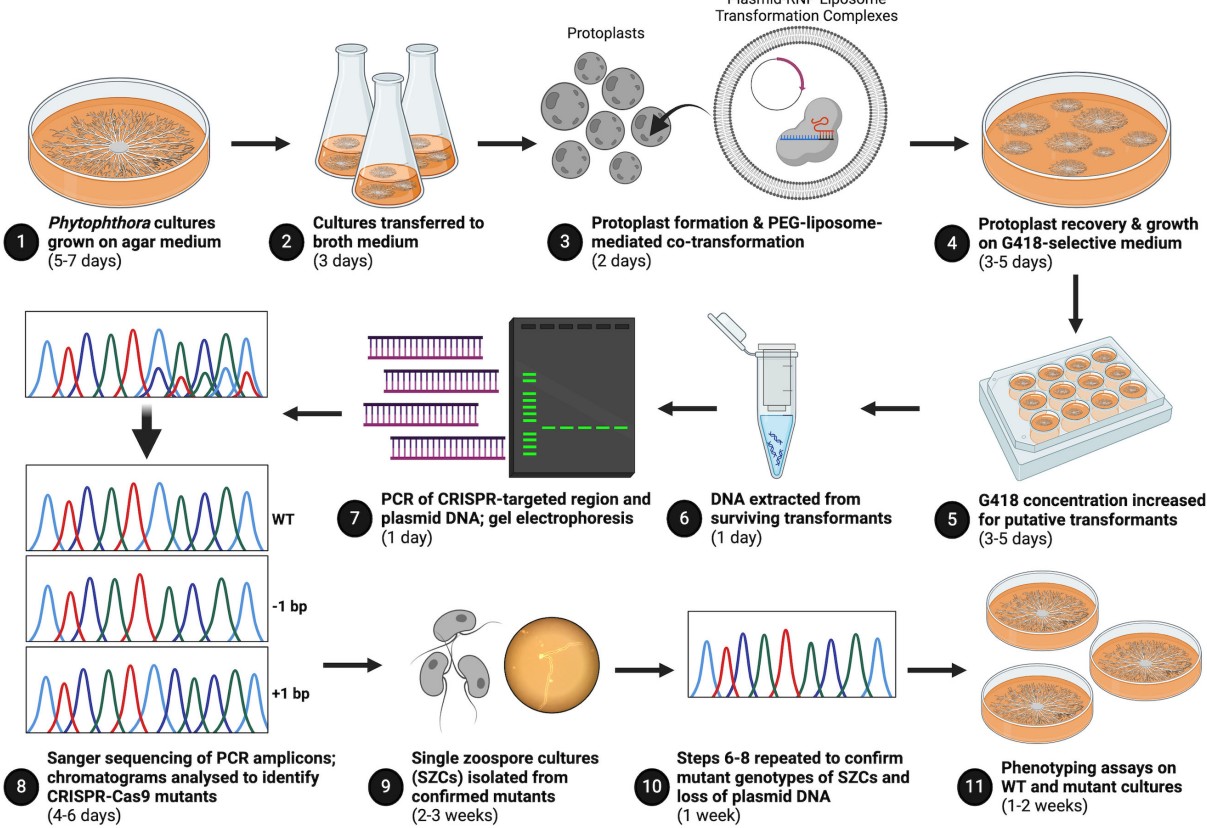

**FIG 2** Method/workflow developed for CRISPR-Cas9 gene editing in *Phytophthora cactorum* using a plasmid-RNP co-transformation protocol. Created in BioRender (Dort, 2023; https://BioRender.com/g46r592).

the PT9 and PT10 indels lead to frameshift mutations in the *PcaORP1* exons and therefore likely result in non-functional proteins translated from the mutated allele (Fig. S5).

### *P. ramorum*

We were unable to obtain any CRISPR-Cas9 mutants in *P. ramorum* NA2_17 using two different plasmid-RNP co-transformation protocols and four unique gRNAs (Data S1). The first protocol we tested in *P. ramorum* was based on a previous method developed to edit the genome of the oomycete *A. invadans* by delivering CRISPR-Cas9 RNPs via a PEG-mediated transformation of protoplasts (32). However, we did not obtain any G418-resistant cultures, indicating that this transformation approach did not work for our *P. ramorum* protoplasts. To test the possibility that the plasmid-RNP complexes had been successfully delivered to *P. ramorum* protoplasts, but that the transformants were simply not expressing the *nptII* gene from the pYF2-PsCG plasmid, we screened a selection of cultures that we had transformed but grown on non-selective recovery medium. The Sanger sequencing results from the screened cultures (representing all four gRNAs tested) indicated that none were CRISPR-Cas9 mutants. We also tested the protoplast transformation protocol developed for CRISPR-Cas9 plasmid delivery in *P. sojae* (65) to co-transform *P. ramorum* NA2_17 with plasmid-RNP complexes, but again, did not obtain any G418-resistant transformants.

### SZCs of *P. cactorum* mutants lose plasmid DNA

The results from PCR performed on the original PT9 and PT10 mutant cultures and their respective SZCs showed loss of pYF2-PsCG plasmid DNA in the single zoospore-derived cultures (Fig. S6). An exception was the culture PT10-1 C2, which was one of two replicate

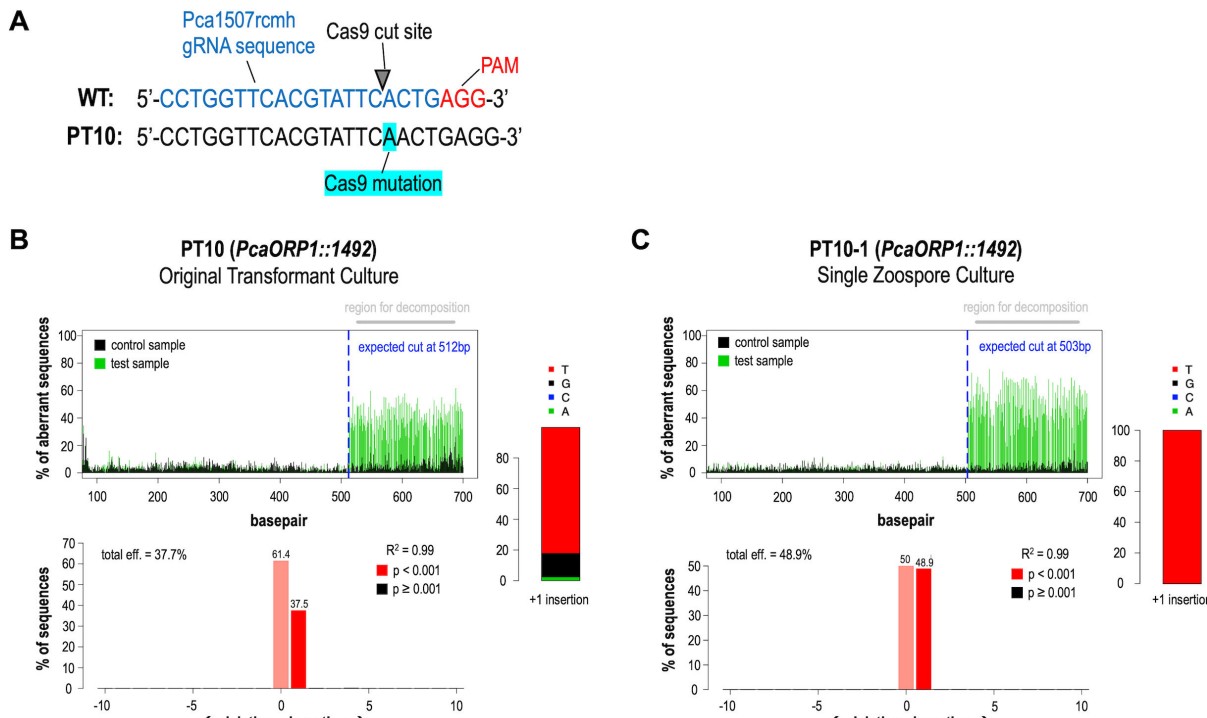

**FIG 3** Details of the CRISPR-Cas9 *PcaORP1::1492* indel mutant, PT10, obtained in *Phytophthora cactorum* FF42 using a plasmid-RNP co-transformation method with the Pca1507rcmh guide RNA (gRNA). (A) The PT10 mutant is heterozygous, with the mutant allele comprising a single base pair insertion relative to the wild-type (WT) sequence. The insertion occurred at the Cas9 cut site, three base pairs upstream of the protospacer adjacent motif (PAM). The graphs shown in panels (B) and (C) were generated using TIDE software (70) on the forward Sanger sequencing chromatograms generated from PcaORPh_Cas9Edit_1F/1R PCR amplicons. The top graph shows the chromatogram sequence compositions of WT (black) versus mutant (green) DNA with the expected Cas9 cut site indicated by the dashed blue line. The bottom graph shows the predicted percentage of sequences in the DNA extraction with WT versus mutant genotypes, and the bar graph to the right indicates the predicted nucleotide in the insertion mutants (*y*-axis represents probability). (B) TIDE decomposition graphs for the original PT10 transformant culture recovered from G418-selective medium. (C) TIDE decomposition graphs for PT10-1, the first single zoospore-derived culture (SZC) generated from PT10. The TIDE results from the other two SZCs tested (PT10-2, PT10-3) were nearly identical (Fig. S3).

cultures from the original PT10-1 SZC (C1 and C2). The DNA from PT10-1 C2 produced a slightly larger than expected PCR amplicon (Fig. S6). To test whether the product was due to non-specific primer amplification, the amplicon was sent for Sanger sequencing; however, the results indicated that the sequence of the PT10-1 C2 PCR product was identical to the products from both the purified pYF2-PsCG DNA and the original PT10 mutant culture DNA. It was therefore assumed that the PT10-1 C2 culture contained plasmid DNA, and the culture was not used for any downstream phenotype assays.

## *PcaORP1* mutant cultures exhibit decreased resistance to oxathiapiprolin

We selected two independent SZCs from each of the WT, PT9, and PT10 cultures for oxathiapiprolin growth assays: WT-1, WT-2, PT9-2, PT9-3, PT10-1 (C1), and PT10-3 (C1). To account for random variation in radial growth, the results from the two SZCs of each culture were pooled prior to statistical analysis (for a total of eight replicates per isolate per treatment) so that changes in growth were ultimately measured between three groups: WT, PT9, and PT10. The raw growth results from the oxathiapiprolin assays are reported in Data S3. Radial growth of both the PT9 and PT10 *PcaORP1* mutant cultures was visibly reduced relative to the WT cultures on oxathiapiprolin-supplemented medium (Fig. 5), and this reduction in growth was statistically significant at every oxathiapiprolin concentration tested (Fig. 6; Data S3). The 0.01, 0.1, and 1.0 µg/mL oxathiapiprolin treatments were not included in the analyses because there was not enough mycelial growth on the V8A plates (Data S3).

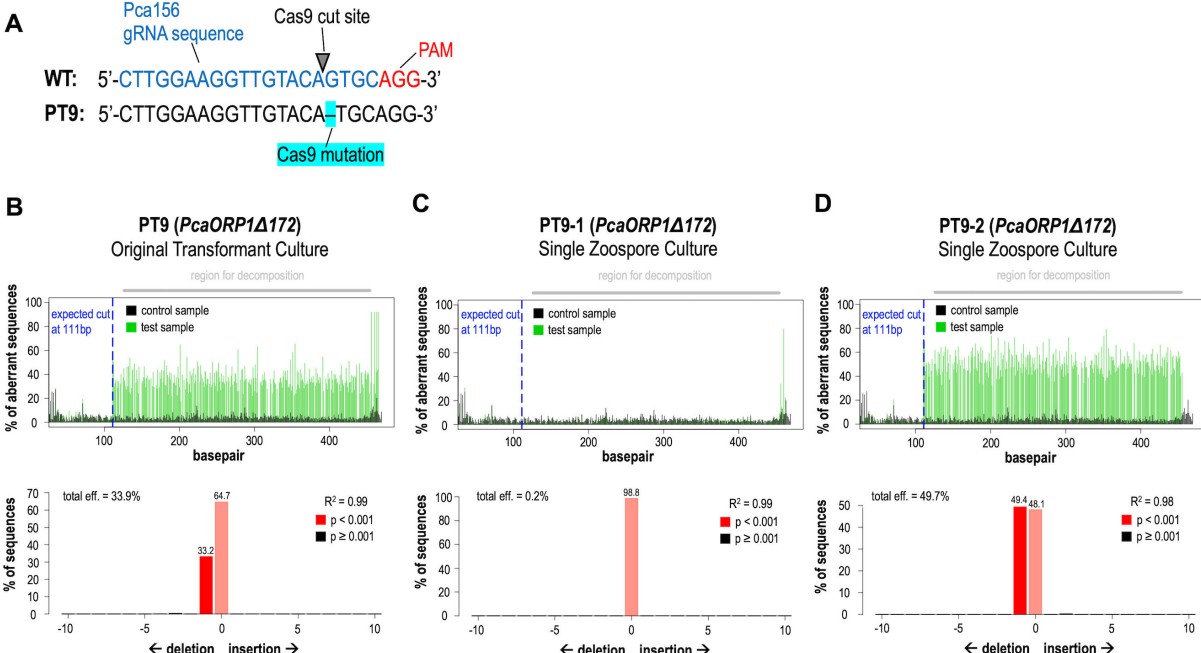

**FIG 4** Details of the CRISPR-Cas9 *PcaORP1Δ172* indel mutant, PT9, obtained in *Phytophthora cactorum* FF42 using a plasmid-RNP co-transformation method with the Pca156 guide RNA (gRNA). (A) The PT9 mutant is heterozygous, with the mutant allele comprising a single base pair deletion relative to the wild-type (WT) sequence. The deletion occurred at the Cas9 cut site, three base pairs upstream of the protospacer adjacent motif (PAM). The graphs shown in panels (B), (C), and (D) were generated using TIDE software (70) on the forward Sanger sequencing chromatograms generated from PcaORPh_Cas9Edit_4F/4R PCR amplicons. The top graph shows the chromatogram sequence compositions of WT (black) versus mutant (green) DNA with the expected Cas9 cut site indicated by the dashed blue line, and the bottom graph shows the predicted percentage of sequences in the DNA extraction with WT versus mutant genotypes. (A) TIDE decomposition graphs for the original PT9 transformant culture recovered from G418-selective medium. (B) TIDE decomposition graphs for PT9-1, the first single zoospore-derived cultures (SZC) generated from PT9. The TIDE results indicated that PT9-1 was derived from a non-edited WT nucleus as it did not have any mutant alleles. (C) TIDE decomposition graphs for PT9-2, the second SZC generated from PT9. The TIDE results from the other two SZCs tested (PT9-3, PT9-4) were nearly identical (Fig. S4).

## DISCUSSION

Using a plasmid-RNP co-transformation method, we successfully edited the *ORP1* gene of *P. cactorum* (*PcaORP1*), obtaining two, independent heterozygous indel mutants. To our knowledge, this is the first successful use of RNPs for CRISPR-Cas9 gene editing in a *Phytophthora* species, and the first implementation of a CRISPR-Cas9 gene editing system in a forest *Phytophthora* pathogen. Delivering both the plasmid selective marker and the RNP in a liposome complex allowed us to efficiently screen for putative CRISPR-Cas9 mutants from G418-resistant transformants given the lack of selective phenotype for any *PcaORP1* mutants. Our results with this plasmid-RNP co-transformation approach, both the success we achieved in *P. cactorum* as well as our inability to obtain *P. ramorum* mutants, provide foundational knowledge for the use of CRISPR-Cas9 RNPs in forest Phytophthoras and give insight into potential future directions for this research.

The two mutants we obtained in *P. cactorum* were the result of experiments with two gRNAs targeting different regions of the *PcaORP1* gene (Pca156, Pca1507rcmh). While we were able to obtain G418-resistant transformants from the co-transformations with the other three gRNAs that we designed (Pca330, Pca1272rcmh, Pca2248mh), none of the transformants were *PcaORP1* mutants. This variability in gRNA success is consistent with previous studies. For example, in 2018, Majeed et al. edited the genome of the oomycete *A. invadans* using RNPs with fluorescently labeled tracrRNA, and while they confirmed successful RNP delivery by observing fluorescence in all RNP-treated protoplasts, they also found that only one of the three gRNAs they tested produced successful CRISPR-Cas9 mutants (32). It is also well-established in many other eukaryotic organisms that

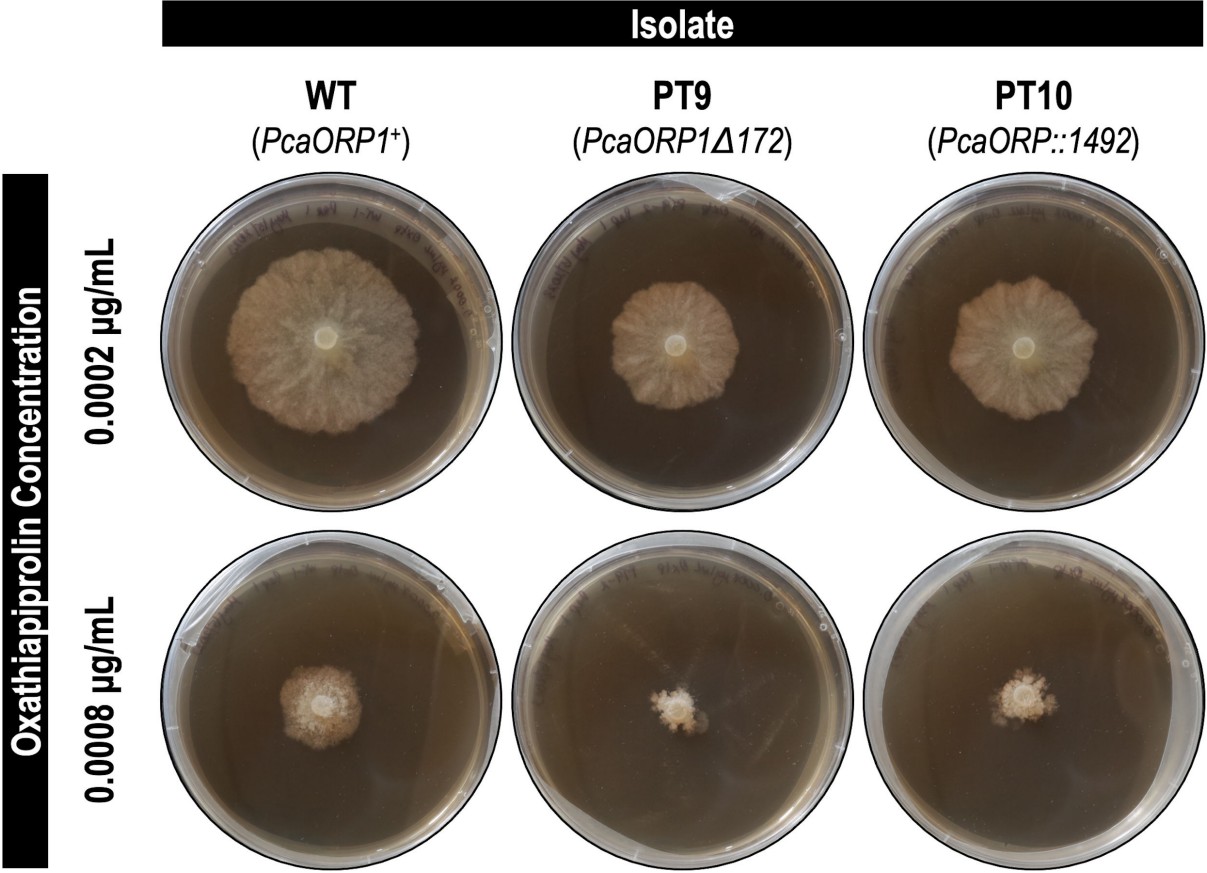

**FIG 5** Photos of *Phytophthora cactorum* FF42 wild-type (WT) and CRISPR-Cas9 *PcaORP1* mutant (PT9, PT10) cultures growing on oxathiapiprolin-supplemented V8 agar 7 days post-plating. The images shown are only a representation of the isolates and oxathiapiprolin concentrations tested; however, the same pattern of reduced growth in the *PcaORP1* mutant cultures was observed at every concentration tested (see Fig. 6).

gRNAs exhibit highly variable activity even when well-designed, and there are many complex factors that can contribute to the success of a gRNA in directing Cas9 to cleave target DNA (79–84). Previous research in human cells has shown significant variability in CRISPR-Cas9 editing performance even between gRNAs designed to target adjacent sites of the same exon (85). Therefore, while our Pca330, Pca1272rcmh, and Pca2248mh gRNAs had appropriate efficiency scores and minimal secondary structures, it is possible they did not perform well with the Cas9 nuclease once transformed into *P. cactorum* cells. These results reinforce the importance of designing and testing multiple gRNAs when developing CRISPR-Cas9 protocols.

The loss of pYF2-PsCG plasmid DNA in most of the single zoospore-derived cultures (SZCs) of the two *PcaORP1* mutants indicates that the plasmid likely remained as extrachromosomal DNA in transformed *P. cactorum* cells rather than integrating into the genome. The fate of plasmid DNA post-transformation is variable in oomycetes. Early studies in *Phytophthora* and *Saprolegnia* species demonstrated that PEG-mediated transformations generally resulted in plasmid DNA being integrated into the genome either at one or multiple sites, and often in tandem repeats (86–89). Due to the randomness of plasmid integration sites in oomycete genomes after transformation, it is hypothesized that this process does not involve homologous recombination, but likely occurs via nonhomologous chromosomal integration (90). However, plasmid integration appears to be species-specific; for example, transformation studies in *P. parasitica* demonstrate that plasmid DNA is often not integrated into the genome, but rather remains extrachromosomal, often leading to instability and loss of antibiotic resistance after subculturing on non-selective media (91). In another oomycete, *Achlya*

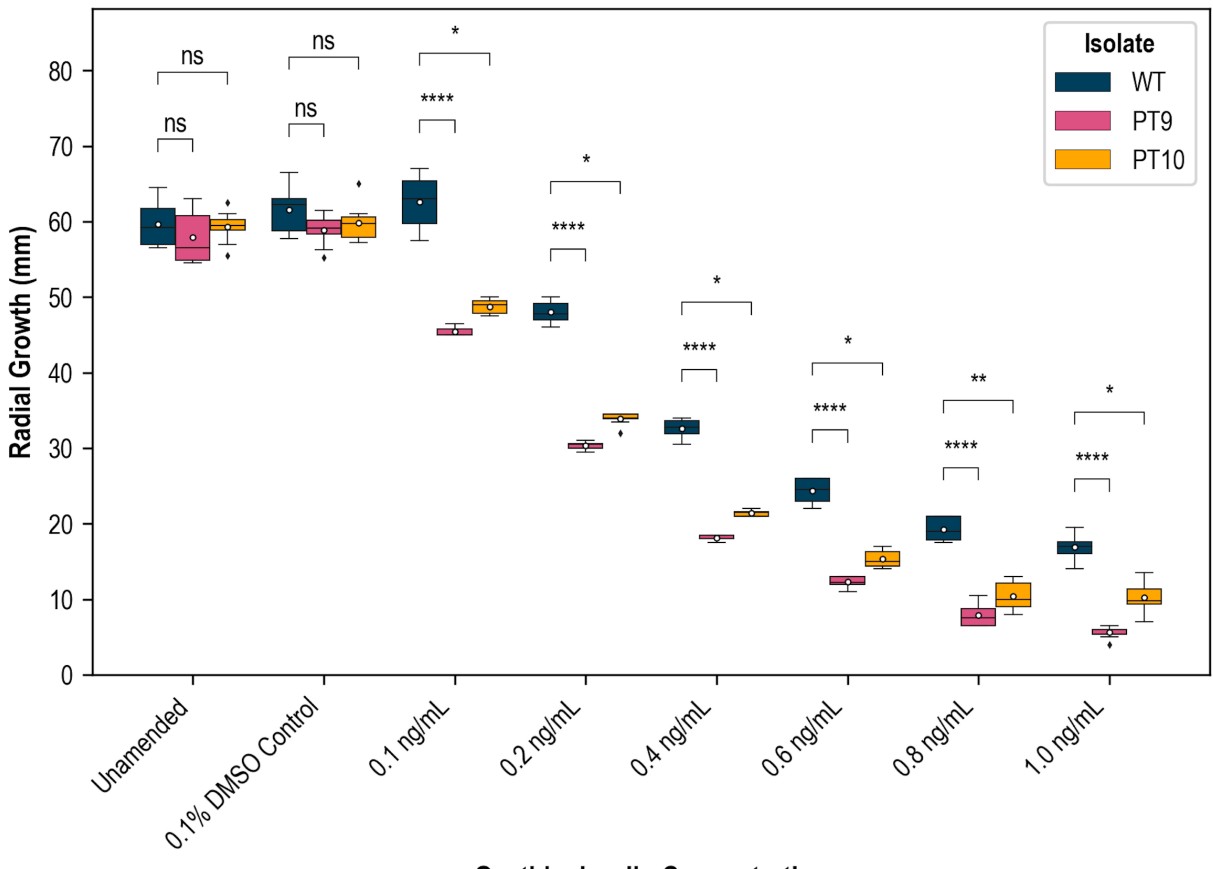

**FIG 6** Boxplot of radial growth measurements of *Phytophthora cactorum* FF42 wild-type (WT) and CRISPR-Cas9 *PcaORP1* mutant (PT9, PT10) cultures growing on V8 agar (V8A) supplemented with a gradient of oxathiapiprolin concentrations (note that the concentrations were converted from µg/mL to ng/mL for this plot). Unamended V8A and V8A + 0.1% DMSO were used as control treatments. The white circles indicate the mean growth value for each isolate at each treatment, and the asterisks represent statistical significance ("ns": not significant, *$P < 0.05$, **$P < 0.01$, ***$P < 0.001$, ****$P < 0.0001$).

*ambisexualis*, it was demonstrated that plasmid DNA was both integrated and maintained episomally after PEG-mediated transformation (92), further demonstrating the diversity in cellular responses to plasmid DNA transformation in oomycetes. As the pYF2-PsCG plasmid does not contain an autonomously replicating sequence (ARS), it would not be maintained episomally in our *PcaORP1* mutants. Instead, the plasmid was likely partitioned away during mitosis (93), resulting in daughter nuclei and subsequently zoospores with no pYF2-PsCG DNA.

While our success in obtaining two *PcaORP1* mutants indicates that RNP-mediated CRISPR-Cas9 gene editing is feasible in *P. cactorum*, the rates of editing we achieved were low. We screened a total of 47 transformants and found two mutants, which translates to a rate of approximately 4%. One possible explanation for this low mutation rate could be the efficiency of the gRNAs, which was previously discussed. Future studies could test more gRNAs in different regions of the *PcaORP1* gene to determine whether there is any effect on the number of mutants recovered. Another possible explanation for the low number of mutants recovered could be linked to the nuclear localization signal (NLS) fused to the Cas9 endonuclease. When developing plasmid-mediated CRISPR-Cas9 gene editing in *P. sojae*, Fang and Tyler found that mammalian NLS sequences were not efficient, and subsequently used a *P. sojae* NLS to target Cas9 to the nuclei of cells (16). The IDT Alt-R S.p. Cas9 nuclease we used has a proprietary NLS of unknown origin; however, it is likely not optimized for oomycete cells given that they are a relatively niche group in CRISPR-Cas9 research. This lack of NLS customization when using a commercially available Cas9 nuclease represents an advantage of a plasmid-mediated

CRISPR-Cas9 gene editing method over an RNP-mediated method. However, our results in *P. cactorum* indicate that the IDT Cas9 NLS is functional in *P. cactorum* and could therefore potentially also be used for other forest Phytophthoras. Future studies could explore optimizing the RNP concentration. Increasing the number of RNP complexes delivered per transformation might increase the likelihood of delivery to nuclei, and therefore the likelihood of successful mutation. Additionally, using fluorescently labeled RNP components, such as tracrRNA, would help in tracking the successful delivery of the RNP to the nuclei and assessing the efficacy of the NLS.

Both *PcaORP1* mutants that we obtained in *P. cactorum* exhibited decreased oxathiapiprolin resistance relative to WT cultures, which is in direct contrast to previous studies of *ORP1* gene mutations in other *Phytophthora* species (11, 21, 26, 55, 59). However, all these previous studies induced specific nonsynonymous point mutations in the ORP1 domain, demonstrating that both heterozygous and homozygous mutations confer increased resistance to oxathiapiprolin (11, 21, 26, 55, 59), likely by altering the binding affinity between the fungicide and ORP1 (26). The heterozygous *PcaORP1* indel mutations we obtained both results in frameshifts, introducing premature stop codons in the PcaORP1 polypeptide encoded by the mutant allele. In theory, this would mean that only one *PcaORP1* allele would be producing functional polypeptides, effectively halving the number of target molecules in the cell to which the ligand (oxathiapiprolin) can bind. Presumably, this would leave a greater concentration of ligand available to target the functional PcaORP1 proteins, leading to a lower protein saturation point. This lower saturation point would be a greater disadvantage for the *PcaORP1* mutants relative to WT individuals at lower concentrations of oxathiapiprolin, which is exactly what we observed.

There are many possible explanations for our lack of success with CRISPR-Cas9 delivery in *P. ramorum*. However, our inability to obtain G418-resistant transformants suggests an issue encountered during the transformation process, which could manifest in one of two ways. The first possible explanation is that the plasmid-RNP complex failed at the point of delivery into the *P. ramorum* protoplasts. *Phytophthora* species, and oomycetes in general, are known to respond variably to transformation protocols (5, 27, 68, 94), so it could be that *P. ramorum* is simply not amenable to the introduction of plasmid-RNP complexes via a PEG-liposome-mediated transformation approach. We did observe that *P. ramorum* consistently produced lower protoplast concentrations than *P. cactorum* with the same protocol, which could have affected transformation success. Releasing protoplasts from hyphae with the quality and concentration sufficient for successful transformation is known to be challenging for oomycetes (90). Additionally, a previous study optimizing transformation protocols in oomycetes found that *Phytophthora* species with higher protoplast concentrations produced more transformants (68), which lends support to the idea that the lower protoplast concentrations we obtained for *P. ramorum* in this study may have contributed to the lack of transformation, and therefore gene editing, success. The second possible explanation is that the RNP and plasmid were delivered successfully, but their presence in the cell is toxic to *P. ramorum*. The toxicity of Cas9, both in its catalytically active and inactive forms, has been previously demonstrated in eukaryotic organisms (35, 36, 95), including *Phytophthora infestans* (38, 39). Given that in our transformations, we delivered Cas9 in both the pYF2-PsCG plasmid and the purified RNP complex, it is possible that its presence in *P. ramorum* cells had toxic effects. However, the fact that we could recover apparently healthy, transformed cultures on non-G418 medium suggests that the plasmid-RNP components were not toxic, but rather were not being delivered to and/or expressed in the *P. ramorum* cells. These results point to the need to establish an improved transformation method for *P. ramorum*, including optimizing protoplast release before further CRISPR-Cas9 experiments are attempted.

Another factor to consider for future CRISPR-Cas9 protocol development in *P. ramorum* is the concentration of RNP delivered to protoplasts during transformation. In our experiments with *P. ramorum*, we delivered roughly equimolar amounts of plasmid

DNA and RNP, testing co-transformations with 0.75 and 1.5 pmol of both components. We selected these RNP amounts based on a combination of the previously developed RNP-mediated CRISPR-Cas9 editing protocol in *A. invadans* (32) and the picomoles of RNP per transformation used in the IDT protocol for cationic lipid delivery of CRISPR RNP complexes (69). However, in *P. cactorum*, we shifted to a higher molar ratio of RNP to plasmid, ultimately using 75 pmol of RNP and 4 pmol of plasmid DNA for each transformation. Increasing the RNP concentration per transformation should therefore be tested in any future studies developing RNP-mediated CRISPR-Cas9 editing in *P. ramorum* once a more consistent transformation protocol has been established.

Developing CRISPR-Cas9 gene editing in forest Phytophthoras is an important step for continued exploration and elucidation of the specific molecular mechanisms driving the phytopathogenic behaviors of these species. Our results provide a proof of concept for the use of RNP-mediated CRISPR-Cas9 gene editing in *P. cactorum*, and our findings serve as a foundation for future studies developing CRISPR-Cas9 protocols in *P. ramorum* and other forest Phytophthoras.

## ACKNOWLEDGMENTS

The authors would like to thank Dr. Felipe Arredondo and Dr. Guillaume Bilodeau, who kindly provided plasmid DNA and Phytophthora cultures for our study, respectively. Thank you also to Dr. Phil Tanguay, who provided experimental advice for co-transformations. The authors would also like to acknowledge FPInnovations—and specifically thank Stacey Kus, Dr. Angela Dale, and Dr. Rod Stirling—for providing research support and access to a PPC-2 facility for our work with *Phytophthora ramorum*. Finally, the authors would like to thank Dr. Monica Gerth, Dr. Guus Bakkeren, and Dr. Debra Wertman for their suggested edits.

This work was supported by a Genome Canada Large-Scale Applied Research Project (LSARP) under Grant #10106 (bioSAFE: Biosurveillance of Alien Forest Enemies) with additional funding from Genome B.C., Genome Québec, Canadian Food Inspection Agency, Natural Resources Canada, and FPInnovations. The authors also acknowledge the support of the Natural Sciences and Engineering Research Council of Canada (NSERC CGS-D).

## AUTHOR AFFILIATIONS

[1]Department of Forest & Conservation Sciences, Faculty of Forestry, University of British Columbia, Vancouver, British Columbia, Canada

[2]Pacific Forestry Centre, Canadian Forest Service, Natural Resources Canada, Victoria, British Columbia, Canada

[3]Institut de Biologie Intégrative et des Systèmes (IBIS), Université Laval, Québec, Quebec, Canada

[4]Département des Sciences du bois et de la Forêt, Faculté de Foresterie et Géographie, Université Laval, Québec, Quebec, Canada

## AUTHOR ORCIDs

Erika N. Dort  http://orcid.org/0000-0002-7814-1060
Nicolas Feau  http://orcid.org/0000-0001-5925-9867

## FUNDING

| Funder | Grant(s) | Author(s) |
|---|---|---|
| Genome Canada (GC) | 10106 | Richard C. Hamelin |
| Canadian Government \| Natural Sciences and Engineering Research Council of Canada (NSERC) | CGS-D | Erika N. Dort |

## AUTHOR CONTRIBUTIONS

Erika N. Dort, Conceptualization, Data curation, Formal analysis, Funding acquisition, Investigation, Methodology, Project administration, Resources, Validation, Visualization, Writing – original draft, Writing – review and editing | Nicolas Feau, Conceptualization, Funding acquisition, Methodology, Supervision, Writing – review and editing | Richard C. Hamelin, Conceptualization, Funding acquisition, Methodology, Resources, Supervision, Writing – review and editing

## ADDITIONAL FILES

The following material is available online.

### Supplemental Material

**Data S1 (Spectrum03012-24-s0001.xlsx).** Summary of plasmid-RNP co-transformations tested in *Phytophthora ramorum* and *Phytophthora cactorum*.
**Data S2 (Spectrum03012-24-s0002.txt).** DNA and amino acid sequences for PrORP1 and PcaORP1.
**Data S3 (Spectrum03012-24-s0003.xlsx).** Culture growth data from oxathiapiprolin assay experiments and results of the statistical tests performed on those growth data.
**File S1 (Spectrum03012-24-s0004.pdf).** Supplemental methods, tables, and figures.

### Open Peer Review

**PEER REVIEW HISTORY (review-history.pdf).** An accounting of the reviewer comments and feedback.

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
