## [Reviewer comments · Microbiology Spectrum]

Microbiology Spectrum

Novel application of ribonucleoprotein-mediated CRISPR-Cas9 gene editing in plant pathogenic oomycete species

Erika Dort, Nicolas Feau, and Richard Hamelin

Corresponding Author(s): Erika Dort, The University of British Columbia

Review Timeline:

Submission Date:	November 25, 2024
Editorial Decision:	December 20, 2024
Revision Received:	January 28, 2025
Accepted:	January 30, 2025

Editor: Lindsey Burbank

Reviewer(s): The reviewers have opted to remain anonymous.

Transaction Report:

DOI: <https://doi.org/10.1128/spectrum.03012-24>

Re: Spectrum03012-24 (Novel application of ribonucleoprotein-mediated CRISPR-Cas9 gene editing in plant pathogenic oomycete species)

Dear Dr. Erika N Dort:

Thank you for the privilege of reviewing your work. Below you will find my comments, instructions from the Spectrum editorial office, and the reviewer comments.

Revision Guidelines

Sincerely,
Lindsey Burbank
Editor
Microbiology Spectrum

Editor comments: The reviewers feedback is outlined below. Some suggestions are made to improve clarity, and enhance the discussion related to previous work. In this particular case since this paper is intended to describe development of a method, I do not have an issue with the methods being the longest part of the manuscript. However, it might be worth considering if there is some content that would be better fit in the supplemental materials to make it a faster read. Also note that novelty is not a requirement for acceptance at Spectrum.

Reviewer #1 (Comments for the Author):

The present study used the commercialized Cas9 protein to combine with sgRNA in vitro to form a Cas9-sgRNA complex, and then assembled it into liposome nanoparticles. In 2016, Fang et al. developed a CRISPR/Cas9-based gene editing method, which has been widely used in a variety of oomycete species. The main difference between this study and Fang's method is that Fang used a plasmid to express Cas9 protein in the cell, rather than directly transferring the Cas9 protein.

I have a few questions about this study.

1. Fang's CRISPR/Cas9 method is applicable to many Phytophthora species, but not to *P. cactorum* and *P. ramorum*? Did the author try the method? What is the significance of this study?
2. In the study, ORP1, the target of the fungicide oxathiapiprolin, was selected to test the method. However, only two transformants were obtained, one with a base insertion and the other with a base deletion. The ORP1 protein was both frameshifted. Why didn't the author site-directly mutate the sites (e.g. S768Y, L863W) conferring oxathiapiprolin resistance? If so, oxathiapiprolin could be used as a selection marker, and the G418 selection marker in PYF plasmid is no longer needed.
3. The author designed 5 sgRNAs for *P. cactorum*, but only 2 successfully mediated the gene editing. Does this mean that the editing efficiency of this method is not high?

Reviewer #2 (Comments for the Author):

In this manuscript, the authors report the first successful use of RNPs for CRISPR-Cas9 gene editing in a Phytophthora species. Techniques for gene editing Phytophthora species lag behind those of other organisms (and it is notoriously difficult). Therefore, this is a very useful addition to the field - and will facilitate future studies in other Phytophthora species. It is also useful that the authors reported what didn't work for one species.

Overall, the manuscript is clear, the techniques appropriate, and the data appears sound. I have only minor suggested revisions:

Table 1. It is very helpful that the EuPaGDT scores are explained in the methods (i.e., a good score is anything above 0.5). It would be nice to add a similar explanation for the IDT gRNA scores. The text mentions that only the highest scores and minimal secondary structure were ordered, but that didn't really help me interpret the scores in Table 1. Is there a similar cutoff for the IDT score (e.g. a 'good' score is anything above..?)

Figures 4-5. Please increase the font size of the figure text (too small to read). Alternatively, delete the text if it doesn't need to be read.

Figure 6 - please add the expected PCR product size (bp) in the figure legend.

Discussion page 29. Were there any issues with obtaining *P. ramorum* protoplasts? It appears from S1 that substantially lower amounts of protoplasts were obtained/used as compared to the successful *P. cactorum* transformations. Optional suggestion, but it might be worth discussing a bit further.

Reviewer #3 (Comments for the Author):

The manuscript by Dort and colleagues investigates the use of RNPs to generate KO mutants in oomycetes, focusing on tree-infecting Phytophthora species. The findings are interesting and timely. The authors focus on neglected Phytophthora species that deserve better attention. Please find below some comments to improve the manuscript.

- The Material and Methods section is very detailed. I understand the authors want to make sure their protocol can be reused. This is appreciated. However, a better strategy may be to provide a supplementary protocol containing details. The Material and Methods section of the main manuscript could then be kept to a reasonable length. Currently, this section has more than 3000 words, while Introduction and Discussion are both in the range of 1000 words. Said differently, half of the manuscript is about technical considerations, which somehow impairs readability.

- The Results section gives a lot of details about sgRNA naming (e.g., "The four PrORP1 gRNAs were named Pr108rc, Pr192, Pr368mh, and Pr751rc" lines 457-458), but it is not clear why these names were selected. Does "rc" stand for "reverse complement"? What does "mh" mean? Surprisingly, a whole paragraph is dedicated to naming sgRNAs (and giving EuPaGDT efficiency scores) when the conclusion is that no transformant could be regenerated for *P. ramorum*. The authors could indicate that attempts to transform *P. ramorum* were unsuccessful.

- Figures 1 and 2 are partially redundant
- Figures 3 and 7 are very well-designed and clearly illustrate the main findings
- The graphs in Fig 4 and 5 are too small.
- Figure 6 should be moved to supporting information
- Figure 8: Please change the concentration from $\mu\text{g/ml}$ (I suppose this is what 'ug/ml' means) to ng/ml so the numbers are easier to read.

Dear Dr. Burbank and Microbiology Spectrum Reviewers,

We, the authors of “Novel application of ribonucleoprotein-mediated CRISPR-Cas9 gene editing in plant pathogenic oomycete species”, would like to thank all of you for taking the time to read our manuscript and provide constructive feedback. Below, we have addressed each of the reviewers’ comments point-by-point. Reviewer comments are numbered with our responses following in point form. Any line numbers we reference correspond to the revised manuscript we are submitting (“Marked-Up Manuscript.docx”).

Reviewer #1 Comments:

“The present study used the commercialized Cas9 protein to combine with sgRNA in vitro to form a Cas9-sgRNA complex, and then assembled it into liposome nanoparticles. In 2016, Fang et al. developed a CRISPR/Cas9-based gene editing method, which has been widely used in a variety of oomycete species. The main difference between this study and Fang's method is that Fang used a plasmid to express Cas9 protein in the cell, rather than directly transferring the Cas9 protein. I have a few questions about this study.”

1. *“Fang's CRISPR/Cas9 method is applicable to many *Phytophthora* species, but not to *P. cactorum* and *P. ramorum*? Did the author tried the method? What is the significance of this study?”*
 - In previous experiments, we did test the CRISPR-Cas9 plasmids developed by Fang et al. in several forest *Phytophthora* species, including *P. cactorum* and *P. ramorum* (we recently published the results of that work: Dort and Hamelin, 2024, PLOS ONE). That work was the foundation for this current study; we found that plasmid transformations were very inconsistent between species, and for that reason and several others involving difficulties working with the plasmids in the lab, we decided to pursue the RNP approach to CRISPR-Cas9 gene editing. This approach has been published in the oomycete *Aphanomyces invadans* (Majeed et al., 2018, Parasites & Vectors), however, has never been published in any other oomycetes, including *Phytophthora* species. The significance of this study is that we are the first to establish that an RNP-based CRISPR-Cas9 method works in a *Phytophthora* species, potentially opening up a new pathway for gene editing for *Phytophthora* researchers, especially those struggling with plasmid-based approaches.
2. *“In the study, *ORP1*, the target of the fungicide oxathiapiprolin, was selected to test the method. However, only two transformants were obtained, one with a base insertion and the other with a base deletion. The *ORP1* protein was both frameshifted. Why didn't the author site-directly mutate the sites (e.g. S768Y,*

L863W) conferring oxathiapiprolin resistance? If so, oxathiapiprolin could be used as a selection marker, and the G418 selection marker in PYF plasmid is no longer needed.”

- Unfortunately, because we are only at the initial stages of transformation protocol development in *P. ramorum* and *P. cactorum*, pursuing the NHEJ (non-homologous end-joining) approach to CRISPR-Cas9 editing was the logical first step in establishing a protocol in these species. While it is true that the HDR (homology directed repair) approach would allow for site-directed mutations and the use of oxathiapiprolin as a selection marker, this approach is known to be more complex to establish and was therefore outside the scope of this study. It is absolutely of interest to pursue HDR-mediated editing in future studies, so we hope that it will be established in future work.
3. *“The author designed 5 sgRNAs for P. cactorum, but only 2 successfully mediated the gene editing. Does this mean that the editing efficiency of this method is not high?”*
- Yes, this does mean that the editing efficiency is not high, which we address in the Discussion (L691-711). Our study is meant to act as a proof of concept on which future work can be built to improve CRISPR-Cas9 protocols for *P. cactorum*, *P. ramorum*, and other *Phytophthora* species.

Reviewer #2 Comments:

“In this manuscript, the authors report the first successful use of RNPs for CRISPR-Cas9 gene editing in a Phytophthora species. Techniques for gene editing Phytophthora species lag behind those of other organisms (and it is notoriously difficult). Therefore, this is a very useful addition to the field - and will facilitate future studies in other Phytophthora species. It is also useful that the authors reported what didn't work for one species.

Overall, the manuscript is clear, the techniques appropriate, and the data appears sound. I have only minor suggested revisions.”

1. *“Table 1. It is very helpful that the EuPaGDT scores are explained in the methods (i.e., a good score is anything above 0.5). It would be nice to add a similar explanation for the IDT gRNA scores. The text mentions that only the highest scores and minimal secondary structure were ordered, but that didn't really help me interpret the scores in Table 1. Is there a similar cutoff for the IDT score (e.g. a 'good' score is anything above..?)”*
- Thank you for pointing this out. Unfortunately, IDT does not provide a similar cutoff for their scores; they simply state that a higher score is better (according to their proprietary gRNA on-target model). We have now

edited the corresponding section in our methods (L299-303) to make this process clearer for the reader.

2. *“Figures 4-5 . Please increase the font size of the figure text (too small to read). Alternatively, delete the text if it doesn't need to be read.”*
 - We have now adjusted Figures 4 and 5, as well as Figures S2 and S3, to make the graphs and font larger and easier to read.
3. *“Figure 6 - please add the expected PCR product size (bp) in the figure legend.”*
 - We have now added the product size to the figure legend.
4. *“Discussion page 29. Were there any issues with obtaining P. ramorum protoplasts? It appears from S1 that substantially lower amounts of protoplasts were obtained/used as compared to the successful P. cactorum transformations. Optional suggestion, but it might be worth discussing a bit further.”*
 - This is a good point, thank you. The precedent to this study was another study we performed wherein we tested transformations in several forest *Phytophthora* species, including *P. ramorum* (Dort and Hamelin, 2024, PLOS ONE). What we found in that study is that the amount of protoplasts produced from the same protocol varied between species, but that even when some species produced more than enough protoplasts (according to previous *Phytophthora* transformation studies), it had no effect on transformation success. When we had subsequent discussions with other *Phytophthora* researchers, the consensus seemed to be that a lower number of high quality/viable protoplasts was more important to transformation success than just the concentration. However, we do see the point that in this case, *P. ramorum* did consistently produce fewer protoplasts than *P. cactorum*, and we know this has been discussed in previous oomycete transformation studies, so we have now added a mention of this in the Discussion (L734-750).

Reviewer #3 Comments:

“The manuscript by Dort and colleagues investigates the use of RNPs to generate KO mutants in oomycetes, focusing on tree-infecting *Phytophthora* species. The findings are interesting and timely. The authors focus on neglected *Phytophthora* species that deserve better attention. Please find below some comments to improve the manuscript.”

1. *“The Material and Methods section is very detailed. I understand the authors want to make sure their protocol can be reused. This is appreciated. However, a better strategy may be to provide a supplementary protocol containing details. The Material and Methods section of the main manuscript could then be kept to a reasonable*

length. Currently, this section has more than 3000 words, while Introduction and Discussion are both in the range of 1000 words. Said differently, half of the manuscript is about technical considerations, which somehow impairs readability.”

- We did indeed want to make sure our protocol can be reused, but we also do not want the readability of our paper to be impaired. In response to this feedback, we have now moved several sections of the Materials and Methods to the Supplementary Materials.
2. *“The Results section gives a lot of details about sgRNA naming (e.g., “The four PrORP1 gRNAs were named Pr108rc, Pr192, Pr368mh, and Pr751rc” lines 457-458), but it is not clear why these names were selected. Does “rc” stand for “reverse complement”? What does “mh” mean? Surprisingly, a whole paragraph is dedicated to naming sgRNAs (and giving EuPaGDT efficiency scores) when the conclusion is that no transformant could be regenerated for P. ramorum. The authors could indicate that attempts to transform P. ramorum were unsuccessful.”*
 - Thank you for bringing these clarity issues to our attention. We have now modified this paragraph to be more concise (L510-517), which we hope solves the problem. The naming of the gRNAs is explained in the Table 1 caption. We do indicate that attempts to transform *P. ramorum* were unsuccessful in the Results section following the gRNA paragraph (“CRISPR-Cas9 plasmid-RNP co-transformation produces mutants in *Phytophthora cactorum* but not *Phytophthora ramorum*”)
 3. *“Figures 1 and 2 are partially redundant”*
 - While we think that both Figures 1 and 2 have a place in the results of this study, we understand why Reviewer #3 feels that they are somewhat redundant. Therefore, we have now moved Figure 2 to the Supplementary Materials to account for this.
 4. *“Figures 3 and 7 are very well-designed and clearly illustrate the main findings”*
 - Excellent, thank you!
 5. *“The graphs in Fig 4 and 5 are too small.”*
 - Thank you for this feedback. We have now adjusted Figures 4 and 5, as well as Figures S2 and S3, to make the graphs and font larger and easier to read.
 6. *“Figure 6 should be moved to supporting information”*
 - We have now moved Figure 6 to the Supplementary Materials.

7. *“Figure 8: Please change the concentration from ug/ml (I suppose this is what ‘ug/ml’ means) to ng/ml so the numbers are easier to read.”*

- We appreciate this suggestion - we have now changed the units in Figure 8 to ng/mL.

Author-initiated revisions:

In response to the feedback from Reviewers 2 and 3 that the font and graphs in Figures 4 and 5 were too small, we also revised Figures S2 and S3 accordingly as they are similar figures. We also had to make minor changes to the main text, Supplementary Materials, and figure/table captions to account for changes made in response to reviewer suggestions. All changes that we made are highlighted in the “Marked-Up Manuscript” document we are submitting with the revisions.

Re: Spectrum03012-24R1 (Novel application of ribonucleoprotein-mediated CRISPR-Cas9 gene editing in plant pathogenic oomycete species)

Dear Dr. Erika N Dort:

Your manuscript has been accepted, and I am forwarding it to the ASM production staff for publication. Your paper will first be checked to make sure all elements meet the technical requirements. ASM staff will contact you if anything needs to be revised before copyediting and production can begin. Otherwise, you will be notified when your proofs are ready to be viewed.

Sincerely,
Lindsey Burbank
Editor
Microbiology Spectrum